# Under 10 mortality patterns, risk factors, and mechanisms in low resource settings of Eastern Uganda: An analysis of event history demographic and verbal social autopsy data

Rornald Muhumuza Kananura[1,2]*, Tiziana Leone[1], Tryphena Nareeba[3], Dan Kajungu[3], Peter Waiswa[2,4], Arjan Gjonca[1]

1 Department of International Development, London School of Economics and Political Science, London, United Kingdom, 2 Department of Health Policy Planning and Management, Makerere University School of Public Health, New Mulago Complex, Kampala, Uganda, 3 Makerere University Centre for Health and Population Research (MUCHAP) and Iganga-Mayuge Health and Demographic Surveillance Site, Iganga, Kampala, Uganda, 4 Global Health Department of Public Health Sciences, Karolinska Institutet, Stockholm, Sweden

* r.m.kananura@lse.ac.uk, mk.rornald@musph.ac.ug

**Data Availability Statement:** The third-party data owned by Iganga-Mayuge Health and Demographic Surveillance Site cannot be shared

## Abstract

### Background

Globally, the under-10 years of age mortality has not been comprehensively studied. We applied the life-course perspective in the analysis and interpretation of the event history demographic and verbal autopsy data to examine when and why children die before their 10th birthday.

### Methods

We analysed a decade (2005–2015) of event histories data on 22385 and 1815 verbal autopsies data collected by Iganga-Mayuge HDSS in eastern Uganda. We used the lifetable for mortality estimates and patterns, and Royston-Parmar survival analysis approach for mortality risk factors' assessment.

### Results

The under-10 and 5–9 years of age mortality probabilities were 129 (95% Confidence Interval [CI] = 123–370) per 1000 live births and 11 (95% CI = 7–26) per 1000 children aged 5–9 years, respectively. The top four causes of new-born mortality and stillbirth were antepartum maternal complications (31%), intrapartum-related causes including birth injury, asphyxia and obstructed labour (25%), Low Birth Weight (LBW) and prematurity (20%), and other unidentified perinatal mortality causes (18%). Malaria, protein deficiency including anaemia, diarrhoea or gastrointestinal, and acute respiratory infections were the major causes of mortality among those aged 0–9 years–contributing 88%, 88% and 46% of all causes of mortality for the post-neonatal, child and 5–9 years of age respectively. 33% of all causes of mortality among those aged 5–9 years was a share of Injuries (22%) and gastrointestinal

publicly. Data can be made available to other researchers with approval by the Iganga-Mayuge HDSS research ethics committee. Data access requests may be sent to Collins Gyezayo at info@muchap.mak.ac.ug.

**Funding:** This study is part of the Lead Author's (RMK) PhD project that is funded by the London School of Economics and Political Science. However, the institution had no role in the study design, data collection and analysis, decision to publish, or preparation of the manuscript.

**Competing interests:** The authors have declared that no competing interests exist.

(11%). Regarding the deterministic pattern, nearly 30% of the new-borns and sick children died without access to formal care. Access to the treatment for the top five morbidities was after 4 days of symptoms' recognition. The childhood mortality risk factors were LBW, multiple births, having no partner, adolescence age, rural residence, low education level and belonging to a poor household, but their association was stronger among infants.

## Conclusions

We have identified the vulnerable groups at risk of mortality as LBW children, multiple births, rural dwellers, those whose mother are of low socio-economic position, adolescents and unmarried. The differences in causes of mortalities between children aged 0–5 and 5–9 years were noted. These findings suggest for a strong life-course approach in the design and implementation of child health interventions that target pregnant women and children of all ages.

## Introduction

Whilst there is global enormous evidence on under-five mortality rate [1,2], very little on the level, patterns and mortality causes among 5–10 age group is known as the group is inadequately included in global reporting, yet in countries with high mortality rates, children aged 5–10 may remain at high risk of morbidity and mortality. Globally, the mortality among 5–9 years of age is estimated at 4 per 1000 and out of the 6.22 million children aged 0–14 year that died in 2018, 9% was a share of 5–9 years of age [2]. Furthermore, 60% of the global annual deaths that occur among children aged 5–14 years is a share of those aged 5–9 years [2,3]. In countries with no or poor civil and vital registration system particularly those in sub-Saharan Africa (SSA) region, very little is known on the level, patterns and mortality causes among 5–9 years of age [4]. For instance, based on the recent literature search at the time of this study, there is a dearth of evidence on mortality estimates, causes and risk factors in SSA among those aged 5–9 years. Yet given the high mortality rates in all age groups and many infectious morbidities in SSA, the 5–9 years of age remain susceptible of contracting infections that increase their likelihood of death. For instance, pneumonia, diarrhoea, and malaria that are major causes of under-five mortality recurrently occur among children aged 5–9 years in SSA [5,6] and thus contributing to the prematurity mortality. Additionally, children are always exposed to life-threatening events such as injuries, malnutrition, and non-communicable diseases that may inhibit them from thriving and transforming [7]. The new-born individual risk factors such as Low Birth Weight (LBW) and preterm birth that are associated with the risks of contracting morbidities such as cardiovascular diseases, congenital anomalies, and chronic lung diseases [8–12], may continue affecting their survival beyond infancy and childhood. However, in SSA, there is a dearth of evidence on how such new-born individual factors may affect their survival beyond infancy.

   Further, the effect of family structure such as socio-economic and maternal characteristics affect the birth outcomes and new-born survival at a later age. There is enormous evidence on how family structures and childbirth characteristics affect childhood and adolescence survival in the high-income country (HIC) [13–15] but very little is known for SSA countries. We also know that access to health care is crucial in preventing infection-related mortality causes, but the community and health facility bottlenecks often lead to failures or delays in receiving the

required healthcare services. The persistent healthcare access challenges in SSA has continued to predispose children to inappropriate services such as informal care and certainly compromising the quality of care [16–18]. Assessing when the children with morbidities or at high risk of mortality access treatment in such resources-limited settings is needed to understand the mortality mechanisms.

In settings with expected high childhood mortality, there is a need for information on how the overall system shapes the health of children for them to be able to survive, thrive and transform. In line with the current global Sustainable Development Goal (SDG) theme of "leaving no one behind" [19], United Nations' Secretary-General's Health Global Strategy for Women, children and adolescents [20], and the WHO–UNICEF–Lancet Commission on children [7], evidence that accounts for when and why do children die before their 10th birth day is an initial step for generating optimal solutions that may accelerate the SDG global and national commitments. We, therefore, apply a life-course perspective in the analysis and interpretation of the data to understand how a combination of individual and family factors, morbidity and access to healthcare may be associated with under-10 child mortality. Such a holistic approach is essential in generating prevention and healthcare interventions that not only improve the children survival but also being able to thrive and transform [4,7,21–23]. Therefore, using what is believed to be the largest and most recent event histories and verbal autopsies data in Uganda, we investigate the under-10 child mortality estimates, patterns, mechanisms, and risk factors.

## Material and methods

### Study design and area

The study used 2005–2015 anonymised Iganga-Mayuge Health Demographic and Surveillance Site (HDSS) event histories and Verbal Social Autopsy (VSA) data. The HDSS is in central-eastern Uganda and it collects information on health, household socioeconomic status, and migration. The HDSS covers at least 185 villages in seven sub-counties within the districts of Iganga and Mayuge with at least 100,000 registered population, of which 59% live in rural areas. Details profiling the site have been reported elsewhere [24,25]. These data have not been used for this kind of study before. Table 1 summarises the estimated number of household and population in the two districts. Details about the Iganga-Mayuge HDSS can be found on the site website (http://www.muchap.mak.ac.ug).

### Iganga-Mayuge HDSS data collection procedures

Each household within the HDSS is geo-referenced, and every household member has a unique identifier. The household registration, pregnancy registration, pregnancy outcome and verbal autopsy forms are the main tools used for data collection. The household registration form is used to capture information on household assets, individual status (education level and marital status) and residence status (migration) twice a year. The HDSS uses community health workers known as the village health team to identify and register pregnancies, and pregnancy/birth outcome using HDSS standard registers. The variables collected in these registers include HDSS household ID, expected date of delivery, maternal age, marital status, birth outcome and date, place of birth and sex of the child. The HDSS employees Field Assistants who are responsible for verifying the pregnancy and birth outcome registers. The HDSS uses WHO/INDEPTH Network Verbal Social Autopsy tools to collect causes of death data for every death that occurs [27]. The tools capture VA data for different age groups of the deceased. These include Neonates (0–28 days), Child (29 days to 14 years), and Adult (15 years and above). The HDSS uses WHO ICD 10 coding guidelines [28] for assigning the causes of death and there is a responsible team of trained physicians who assign the causes of death

**Table 1. Estimated number of household and population in Iganga-Mayuge district.**

| County | Subcounty | Household | | Population | | |
|---|---|---|---|---|---|---|
| | | Number | Number Average | Male | Female | Total |
| | **Iganga** | | | | | |
| Bugweri | Busembatia town council | 2,700 | 5.3 | 6,768 | 7,663 | 14,431 |
| | Buyanga | 10,750 | 4.8 | 24,881 | 26,531 | 51,412 |
| | Ibulanku* | 6,753 | 5 | 16,558 | 17,652 | 34,210 |
| | Igombe | 3,793 | 4.5 | 8,223 | 8,952 | 17,175 |
| | Makuutu | 5,203 | 5.4 | 13,490 | 14,502 | 27,992 |
| | Namalemba | 4,461 | 5.3 | 11,426 | 12,221 | 23,647 |
| Municipality | Central division* | 7,827 | 3.8 | 13,673 | 16,718 | 30,391 |
| | Northern division* | 5,975 | 3.9 | 10,717 | 12,762 | 23,479 |
| Kigulu | Bulamogi* | 5,882 | 5 | 14,549 | 15,466 | 30,015 |
| | Nabitende | 5,225 | 5.4 | 13,655 | 14,515 | 28,170 |
| | Nakalama* | 9,167 | 4.9 | 21,485 | 23,649 | 45,134 |
| | Nakigo* | 7,433 | 5.1 | 18,441 | 19,795 | 38,236 |
| | Nambale | 9,403 | 5 | 22,394 | 24,721 | 47,115 |
| | Namungalwe | 7,638 | 4.9 | 17,729 | 19,668 | 37,397 |
| | Nawandala | 5,736 | 5.4 | 15,156 | 16,146 | 31,302 |
| | Nawanyingi | 4,951 | 5.3 | 12,878 | 13,404 | 26,282 |
| | **Mayuge** | | | | | |
| Bunya | Baitambogwe | 8,004 | 4.8 | 18,511 | 19,649 | 38,160 |
| | Bukabooli | 8,527 | 5.2 | 21,413 | 22,841 | 44,254 |
| | Bukatube | 8,156 | 5 | 19,974 | 21,135 | 41,109 |
| | Busakira | 5,545 | 5.5 | 14,723 | 15,566 | 30,289 |
| | Buwaaya* | 4,001 | 5.3 | 10,203 | 11,145 | 21,348 |
| | Imanyiro* | 5,785 | 5.5 | 15,659 | 16,416 | 32,075 |
| | Jaguzi | 3,761 | 3.5 | 6,713 | 6,504 | 13,217 |
| | Kigandalo | 5,854 | 5.4 | 15,427 | 16,200 | 31,627 |
| | Kityerera | 8,598 | 5.5 | 23,100 | 24,220 | 47,320 |
| | Malongo | 22,320 | 4.6 | 50,283 | 52,241 | 102,524 |
| | Mayuge town council | 4,397 | 3.9 | 8,058 | 9,093 | 17,151 |
| | Mpungwe | 4,880 | 5.3 | 12,344 | 13,612 | 25,956 |
| | Wairasa | 7,685 | 4.4 | 16,669 | 17,473 | 34,142 |

*Sub-counties covered by the Iganga-Mayuge HDSS.

Source: 2014 Uganda National Population and Housing Census report [26].

using the ICD-10 classification of diseases. The verbal autopsy is normally conducted after mourning days–usually 3–4 weeks after the event. The data elements under stillbirth and newborn mortality verbal autopsy tool include places of birth, maternal morbidity's experience, childbirth status (live or stillbirth), time the baby died after birth, birth weight, gestation period and causes of death. The data elements for the 29 days-15 years mortality's verbal autopsy tool include the place of treatment before death, the time taken to receive treatment in days, and the causes of death.

In each village, there is a community health worker who notifies and reports all community pregnancies, births, and deaths events within two weeks. Thereafter, the reported events are verified by the DHSS Field Assistants to check for residential status and henceforth use a structured respective standard tool for actual data collection. The HDSS has standard operating

procedures for administering each tool, training, and data management. Refresher training is always conducted before the actual days of data collection and while in the field each team of 4 Field Assistants is assigned to a supervisor whose role is to ensure that the data collection standards are adhered to. The forms that only bare the supervisor signature are then submitted to the Field Manager, who later submits them to the data management office. The data is managed by a team of data scientist who include two statisticians, two computer scientists, one data entry supervisor and four data entrants who make sure that the data is well entered and cleaned.

## Analysis of event histories data for mortality patterns and risk factors

We use the 22,385-event history demographic data records to estimate the age-specific mortality patterns and risk factors. The residents' live births, migration out of the DHSS for less than 6 months and in-migration for at least 6 months were considered for analysis. The HDSS residents are those that have stayed within the site for at least 6 months and those that out-migrated are considered residents for a period of fewer than 6 months before returning to the HDSS. The study main outcome is an event (death) that is experienced after "a live" birth by time t (10th birthday). The study covariates are LBW(<2.5 Kgs)–that is captured from the child immunization card for those that delivered at the health facility, maternal age(grouped as <20 years, 20–29 years and 30 years and above), marital status(1 –having or staying with a partner and 0 –having no partner), household wealth index(least poor for indices 4–5, 3 middle poor and 1–2 poorer), maternal education level, child sex, place of residence, and birth category.

The data was managed and analysed in STATA v.15. The life table approach was used to estimate age-specific mortality patterns (S1 File). To assess the under-10 mortality risk factors, Royston-Parmar flexible parametric model for survival analysis was performed using *stpm2* user-written STATA command [29] with individual-level clustering while controlling for all covariates after multiple imputations. The approach was used because of its flexibility as compared to other survival parametric models such as the exponential and Weibull models [29]. Further, death among children has monotone hazard rate that reduces with time, which is suitable for the Royston-Parmar method. Since the focus was on the hazard rates, the *scale* category considered was a hazard at 3 degrees of freedom. The 3 degree of freedom is the same as a spline with 2 interior knots [29], which provide better estimates [29]. Further, the hazard probability graphs were generated to assess how the mortality varies across the child's age for each of the risk factors. The censoring was done at 120 months (10 years) and the analysis was restricted to children who were born alive.

## Missing data and multiple imputations for event histories data

The extent and pattern of missing data were scrutinised to guide the modelling strategy. Birth weight, wealth index, and Marital status records were missing among 64%, 22% and 20% of the registered birth records respectively (S1 Table). For all other variables, data were missing for <1% of the study participants (S1 Table).

The place of delivery, the time of death, place of residence, education level and maternal age were associated with the birth weight missingness–indicating a possibility of birth weight missingness at random (S2 Table). Similarly, maternal age, marital status and education level were associated with wealth index missingness–indicating the possibility of wealth index missing at random (S2 Table). Marital status missingness was associated with the place of resident and maternal age (S2 Table). Our consideration of missing at random is based on the assumption that the missingness of variables' fields depends on some of the observable variables [30,31].

Given the determinants of missingness, we ran multiple imputations (m = 100) not only controlling for identified factors that contribute to the likelihood of missingness, but also all other variables (as auxiliary) in the dataset. The multiple imputation method has been indicated as an important approach for minimising the missingness bias [32] and this does not depend on the magnitude of missingness [33,34]. There were modest differences between the imputed and complete discriptive statitics for variables with missing field (S1 Table).

## Analysis of VASA data

We use the VASA data on 844 neonatal deaths including stillbirths and 986 children (1 month-10 years) deaths to understand the mortality mechanism by assessing the causes of death and access to the required services at birth and after birth for sick children. The VASA data were provided as a separate file and one of the limitations was that the data could not be linked to the event histories dataset. Based on the discussion with the HDSS data manager, this limitation arises from the fact that during the collection of verbal autopsy data, the collectors were not assigning the VASA form with the standard HDSS household and individual identifiers, which makes it hard to merge the two. Because of the limitation of linking the VASA data to the event histories data, the cause-specific mortality probabilities and rates could not be estimated. Additionally, the dataset did not include all the variables that could be used to assign the causes of death using the available software algorithm such as InSilicoVA [35] or SmartVA [36] or InterVA [37] and thus, the results only depend on the causes of death that were assigned by the Physicians.

# Results

## Event histories study participants information

A total of 22,385 birth were registered between 2005 and 2015 of which 14% were LBW and 3% multiple births (S1 Table). By the end of the follow-up period, the mothers of 4% and 13% of the children were still adolescents and unmarried, respectively. In addition, the mothers of 27% and 47% of the children had no education at all and belonged to a poor(er) household respectively (S2 Table).

## Mortality estimates and patterns

In the estimation of mortality patterns using the life table approach, we adjusted for censoring (Table 2). The under-10 and 5–10 mortality probabilities were 129 per 1000 live births and 11 per 1000 children aged 5–9 years, respectively. Additionally, the probability of infant mortality and under-five mortality was 46 and 100 per 1000 live births, respectively. Overall, the mortality rate per person-years was 47 per 1000 among infants (0–1), 12.2 per 1000 among those aged 1–5 years and 3.1 per 1000 among those aged 5–10 years (Table 2). Comparing the mortality rates by gender, the probability of death and mortality rate per person-years within the different age groups was not different (Table 2).

## Causes of mortality

**Stillbirth and new-born mortality causes.** Of the 844 new-born deaths identified in VASA data, 32.2% were stillbirths. Antepartum and intrapartum complications contributed 56.3% of all mortality causes of stillbirth and new-born death–each contributing 31% and 25% respectively (Table 3). Prematurity and low birth weight contributed almost 20% of all causes of stillbirth and new-born mortality and 18% resulted from other unidentified perinatal causes (Table 3).

**Table 2. Under 10 mortality estimated-abridged life table using 2005–2015 birth registration dataset.**

| Interval | Number of persons at risk ($l_t$) | Deaths ($d_t$) | $C_t$ | $N_t^*$ | $q_t$ (95% Confidence Interval) | $p_t$ | $S_t$ | Person-time (in years) of observation | ($M_t$) | $M_t$ 95% Confidence Interval |
|---|---|---|---|---|---|---|---|---|---|---|
| | | | | | Age-specific mortality | | | | | |
| 0–1 years | 22385 | 946 | 3308 | 20731 | 0.046(0.043–0.131) | 0.954 | 0.954 | 20181.17 | 0.047 | 0.044–0.050 |
| 1–5 years | 18131 | 563 | 11205 | 12528.5 | 0.045(0.041–0.126) | 0.955 | 0.911 | 45971.65 | 0.012 | 0.011–0.014 |
| 5–10 years | 6363 | 38 | 6104 | 3311 | 0.011(0.007–0.026) | 0.989 | 0.901 | 12151.31 | 0.003 | 0.002–0.004 |
| 0–28 days | 22385 | 434 | 0 | 22385 | 0.019(0.017–0.053) | 0.981 | 0.981 | 22385.00 | 0.019 | 0.018–0.021 |
| 0–10 years | 22385 | 1547 | 20835 | 11967.5 | 0.129(0.123–0.370) | 0.871 | 0.871 | 78360.05 | 0.020 | 0.018–0.021 |
| 0–5 years | 22385 | 1509 | 14513 | 15128.5 | 0.100(0.095–0.287) | 0.900 | 0.900 | 66148.51 | 0.023 | 0.021–0.025 |
| | | | | | Male | | | | | - |
| 0–1 years | 11192 | 474 | 1659 | 10362.5 | 0.046(0.042–0.129) | 0.954 | 0.954 | 10088.47 | 0.047 | 0.043–0.051 |
| 1–5 years | 9058 | 296 | 5620 | 6248 | 0.047(0.041–0.129) | 0.953 | 0.909 | 23020.25 | 0.013 | 0.011–0.015 |
| 5–10 years | 3145 | 22 | 3015 | 1637.5 | 0.013(0.008–0.028) | 0.987 | 0.897 | 5991.05 | 0.004 | 0.002–0.006 |
| 0–28 days | 11192 | 211 | 0 | 11192 | 0.019(0.016–0.051) | 0.981 | 0.981 | 11192.00 | 0.019 | 0.016–0.021 |
| 0–10 years | 11192 | 792 | 10399 | 5992.5 | 0.132(0.123–0.374) | 0.868 | 0.868 | 39125.90 | 0.020 | 0.018–0.023 |
| 0–5 years | 11192 | 770 | 7279 | 7552.5 | 0.102(0.095–0.289) | 0.898 | 0.898 | 33120.95 | 0.023 | 0.021–0.026 |
| | | | | | Female | | | | | - |
| 0–1 years | 11193 | 472 | 1649 | 10368.5 | 0.046(0.042–0.128) | 0.954 | 0.954 | 10092.66 | 0.047 | 0.043–0.051 |
| 1–5 years | 9073 | 267 | 5586 | 6280 | 0.043(0.038–0.117) | 0.957 | 0.914 | 22953.03 | 0.012 | 0.009–0.014 |
| 5–10 years | 3218 | 16 | 3085 | 1675.5 | 0.010(0.005–0.020) | 0.990 | 0.905 | 6155.22 | 0.003 | 0.001–0.004 |
| 0–28 days | 11193 | 224 | 0 | 11193 | 0.020(0.017–0.054) | 0.980 | 0.980 | 11193.00 | 0.020 | 0.017–0.023 |
| 0–10 years | 11193 | 755 | 10436 | 5975 | 0.126(0.118–0.356) | 0.874 | 0.874 | 39228.24 | 0.019 | 0.017–0.022 |
| 0–5 years | 11193 | 739 | 7235 | 7575.5 | 0.098(0.091–0.277) | 0.902 | 0.902 | 33040.02 | 0.022 | 0.020–0.025 |

$p_t$–proportion surviving interval t (1- $q_t$), $q_t$–proportion dying during interval t ($d_t$ /$N_t^*$), $S_t$–Survival probability cumulative ($S_{t+1} = p_{t+1}*S_t$), $C_t$–Number of persons lost or Censored, $N_t^*$–Average number participants at interval t adjusted for Censoring ($I_t$-$C_{t/2}$), $M_t$–Person-years mortality proportion ($q_t$ / Person-time).

80% of the mothers experienced morbidity in pregnancy(S3 Table) and these were febrile illness(44%), severe abdominal pain(31%), blurred vision(22%), smelly vaginal discharge (14%), vaginal bleeding (14%), pallor in the last trimester(13%), heart diseases(12%), puffy face (9%), high blood pressure(9%), and convulsion(5%).

**Child (0–10 years) mortality causes.** Malaria, protein deficiency, diarrhoea or gastrointestinal, and acute respiratory infections were major causes of mortality among those aged 0–9 years–contributing 88%, 88% and 46% of all causes of mortality for the 1–12 months (post-neonatal), 1–4 years and 5–9 years of age respectively (Fig 1). Injuries and gastrointestinal contributed 33% of all causes of mortality among those aged 5–9 years (Fig 1).

## Access to health services among children who died

Of the 844 stillbirth and neonatal deaths identified in the VASA data, 63% were delivered in the health facilities and 10% (3.2/33.4%) of community deliveries occurred en-route to the health facilities (Fig 2).

**Table 3. Stillbirth and new-born mortality causes using 2005–2015 Iganga-Mayuge verbal autopsy data.**

| Stillbirth and newborn mortality causes | Stillbirth | | Newborn | | Total | |
|---|---|---|---|---|---|---|
| | Freq. (n = 272) | % | Freq. (n = 572) | % | Freq. (n = 844) | % |
| **Antepartum maternal complications** | | | | | | |
| *Hypertensive disorders* | 34 | 12.5 | 30 | 5.2 | 64 | 7.6 |
| *Other Antepartum Maternal diseases* | 186 | 68.4 | - | - | 186 | 22.0 |
| *Antepartum haemorrhage* | 12 | 4.4 | - | - | 12 | 1.4 |
| **Intrapartum-related causes** | | | | | | |
| *Birth injury and or asphyxia* | - | - | 157 | 27.5 | 157 | 18.6 |
| *Obstructed labour* | 27 | 9.9 | 29 | 5.1 | 56 | 6.6 |
| **Prematurity and or low birth weight** | 13 | 4.8 | 155 | 27.1 | 168 | 19.9 |
| **All other unidentified perinatal causes** | - | - | 149 | 26.1 | 149 | 17.7 |
| **Other causes** | | | | | | |
| Febrile Infections | - | - | 21 | 3.7 | 21 | 2.5 |
| Sepsis or Tetanus | - | - | 18 | 3.2 | 18 | 2.1 |
| Abnormalies | - | - | 9 | 1.6 | 9 | 1.1 |
| Communicable diseases | - | - | 2 | 0.4 | 2 | 0.2 |
| Injuries or accidents | - | - | 2 | 0.4 | 2 | 0.2 |

Of the 986 children that died, 71% of the children accessed treatment from the health facility, of which 20% was from the private health facilities (Fig 2). On average, access to the treatment for the top five morbidities was after 4 days of symptoms' recognition (3 days–Malaria, 3 days–malnutrition, 9 days–protein-energy deficiency, 11 days–Acute respiratory infections, and 3 days—diarrhoea) (Table 4)

## Multivariate analysis for under 10 child mortality risk factors using event histories data

**Association of birth weight and birth category with under-10 mortality.** The under 10 likelihood of death was higher by 53% and 79% among LBW and multiple births respectively (Table 5). The probability of death among LBW was higher among children aged 0–1 years, after which the probability became parallel to that of the normal birth weight up to 3 years but remaining higher among the LBW children (Fig 3). By 5 years, the mortality probability flattened towards zero (Fig 3). Similarly, the mortality probability was higher among multiple children aged 0–1 years, after which the probability became parallel to that of the singleton up to 5 years but remaining higher for multiple birth children (Fig 3). By 6 years, the mortality probability curve flattened towards zero (Fig 3).

**Association of maternal age and marital status with under-10 mortality.** The under-10 mortality likelihood was lower by 32% and 16% among mothers aged 20–29 years and 30 years + respectively relative to those aged less than 20 years (Table 5) and remained parallel until 2 years, beyond which it flattened towards zero (Fig 4). For infants, the mortality likelihood among children whose mothers were adolescents (<20) was extremely high and it remained slightly higher until 3 years (Fig 4). Unmarried women or those with no partners were 22% more likely to experience children mortality within 10 years after birth (Fig 4). The likelihood was higher for infant children beyond which it matched the married counterparts but remaining high until the age of 2 years (Fig 4).

**Association of maternal education and household wealth with under-10 mortality.** The likelihood of under-10 mortality was 19% lower among women with post-primary education level than those who had no education level at all (Table 5). However, the probability was

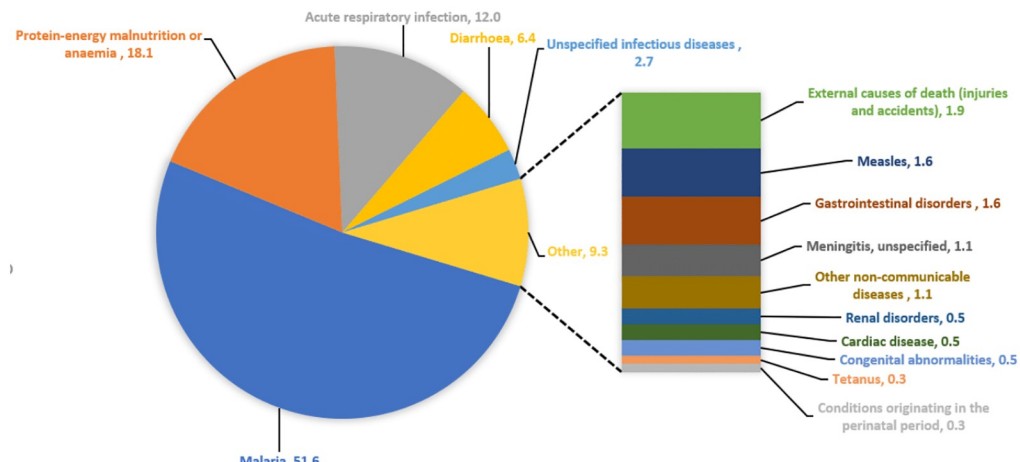

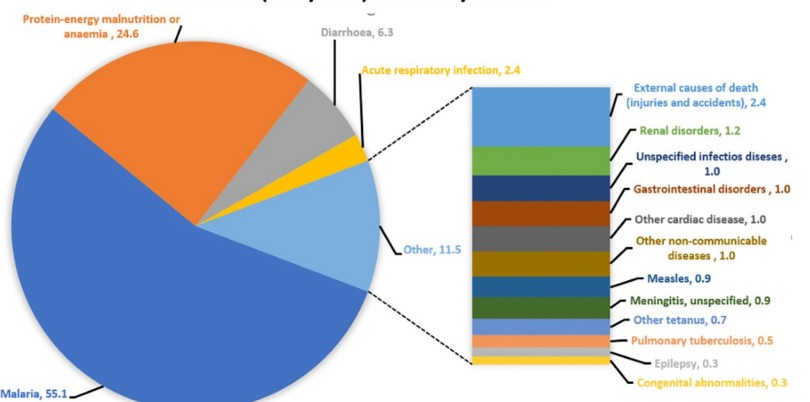

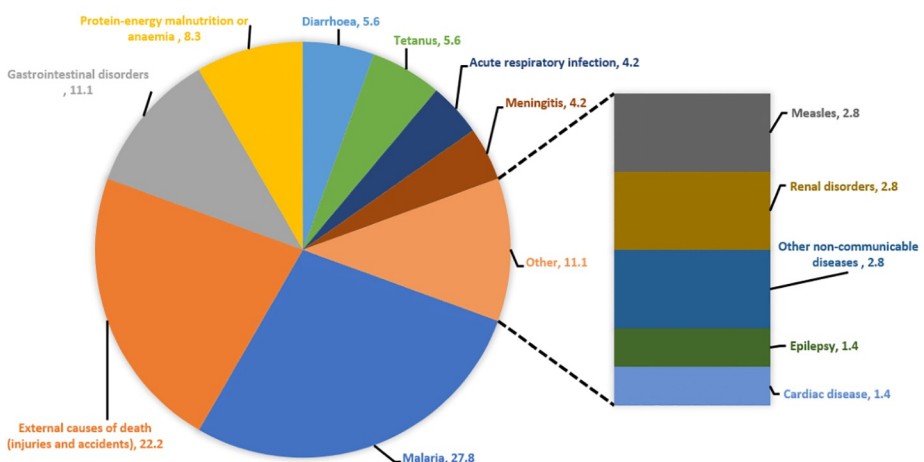

Source: Analysis of 2005-2015 Iganga-Mayuge HDSS Verbal Social Autopsy data.

**Fig 1. Under-10 mortality causes.**

**A. Place of delivery**

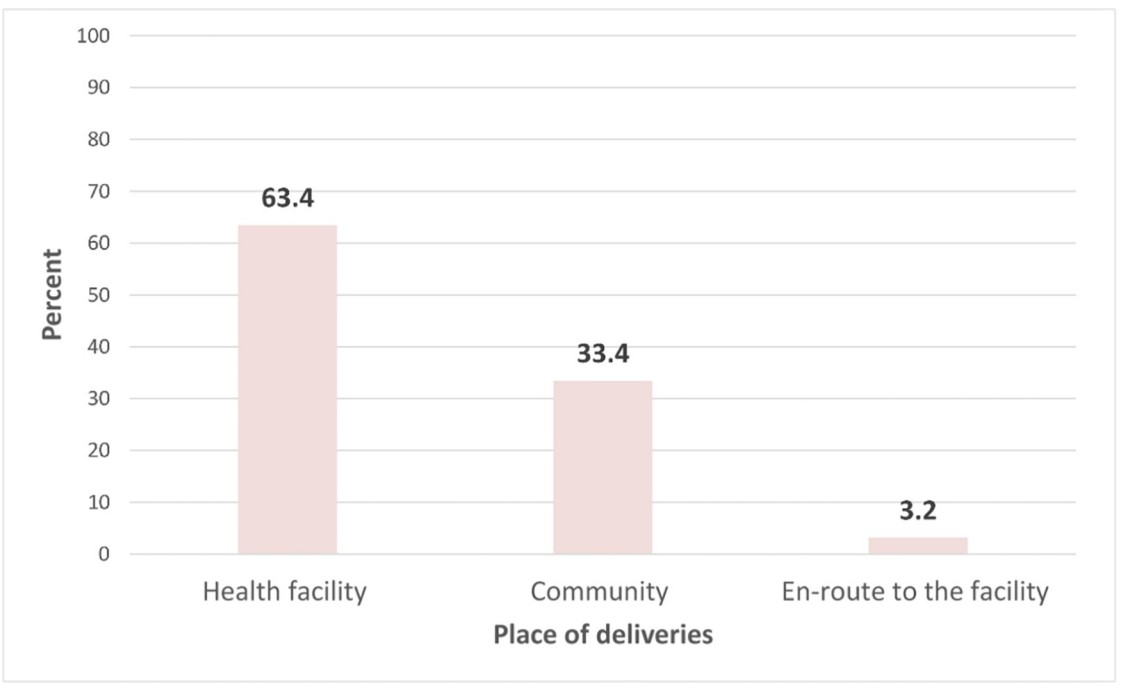

**B. Final place of treatment for sick children**

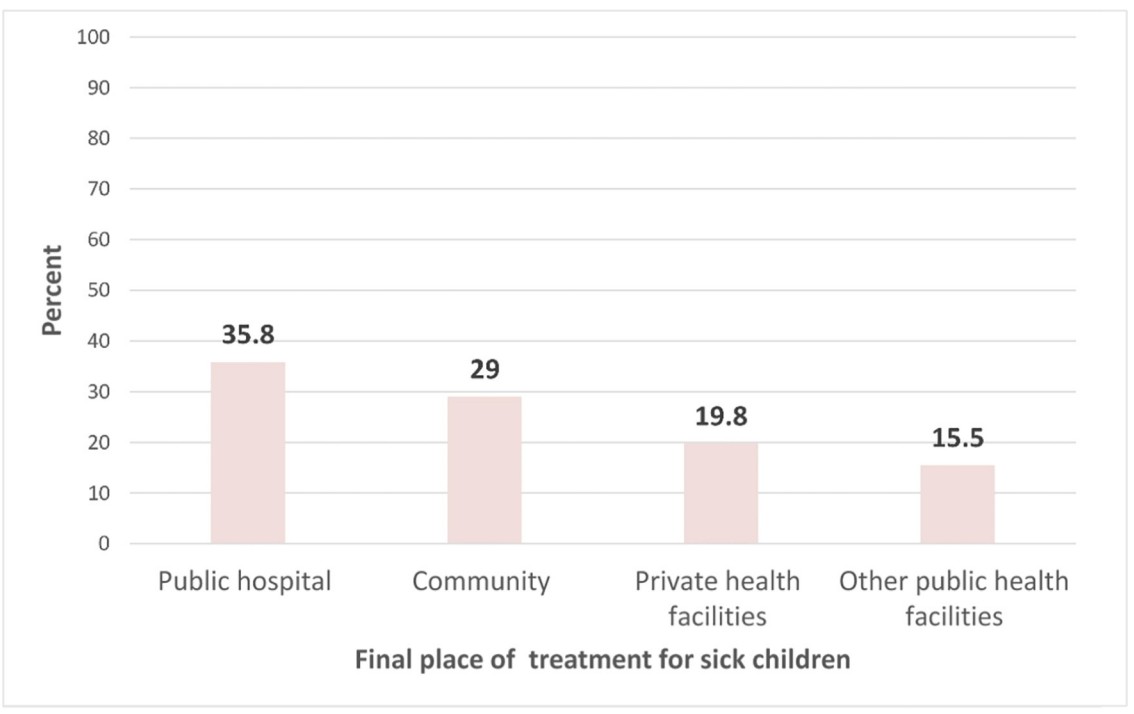

Source: Analysis of 2005-2015 Iganga-Mayuge HDSS Verbal Social Autopsy data.

**Fig 2. Places of birth and treatment for sick children.**

**Table 4. Average number of days before sick children accessed medical treatment after symptom recognition using 2005–2015 Iganga-Mayuge verbal autopsy data.**

| Cause of death | Mean number of the before seeking hospital treatment after symptom recognition |
|---|---:|
| Malaria | 2.4 |
| Malnutrition | 2.7 |
| Severe protein-energy malnutrition | 8.9 |
| Acute respiratory infection | 11.1 |
| Diarrhoea | 3.0 |
| **Total** | **3.9** |

only higher among infant children whose mother's education levels were lower than post-primary, beyond which the likelihood became close to each other but remaining high until 2-years of age and thereafter flattened towards zero at nearly 6 years of age (Fig 5). Similarly, the likelihood of child mortality was 34% lower among the least poor women level relative to those who were poor(er). However, the mortality likelihood was only higher for infants whose mothers were poor(er), beyond which it became parallel but remained lower among the least poor until 2-years of age and flattened toward zero at nearly 6 years of age (Fig 5).

**Relationship between place of residence and under-10 mortality.** The likelihood of mortality was 24% higher among children who were residents of rural communities than their urban counterparts (Table 5). However, the mortality likelihood was also high among urban infants and it remained high in both areas up to the age of 2 years. The mortality probability curve flattened towards zero at the age of 6 years in both areas (Fig 6).

## Discussion

We have attempted to comprehensively provide an insight into when and why children die before their 10th birthday in a resource-limited setting using event history demographic and verbal social autopsy data that was collected by Iganga-Mayuge Demographic Surveillances Site in Eastern Uganda. The results are not entirely new but tend to reaffirm the key findings of previous reports on global and regional mortality. Particularly, our study provides the under-10 and 5–9 years mortality estimates and the importance of HDSS data in mortality estimates in areas with no or weak vital registration systems. Thus, our results contribute to the much-needed accountability of when and why children, particularly in a resource-limited context, die at each critical stage of development. We applied the life-course perspective in the analysis and interpretation of the data to understand how the child individual, maternal, family and community characteristics may be associated with childhood mortality. Similarly, we also assessed mortality mechanism through the analysis of mortality causes and other associated determinants.

The mortality probabilities among under-10 and 5–9 years were 129 per 1000 live births and 11 per 1000 children aged 5–9 years, respectively. Our study's under-five and child mortality probabilities' estimates were slightly higher than the study's regional estimates on ten-year early childhood mortality rates reported in the recent national health survey report [38] and close to the global report on global, regional and national estimates [1,2]. In this study, the mortality probability estimates among 5–9 years of age were close to the reported figures on SSA on the same age group in the global reports [2,3]–indicating a need for investing in Demographic Surveillance Sites for population health monitoring including mortality and death registrations, particularly, in countries with no or poor civil registration and vital data

**Table 5. Imputed Multivariate analysis of child mortality risk factors using 2005–2015 Iganga-Mayuge HDSS event histories data.**

| | Unadjusted | | Adjusted | |
|---|---|---|---|---|
| | HR | 95% Confidence interval | HR | 95% Confidence interval |
| **Birth category** | | | | |
| Singleton | 1.00 | - | 1.00 | - |
| Multiple | 1.76 | 1.37–2.25*** | 1.76 | 1.31–2.35*** |
| **Birth weight** | | | | |
| > = 2.5 Kg | 1.00 | - | 1.00 | - |
| <2.5 Kg | 1.65 | 1.34–2.04*** | 1.75 | 1.33–2.32*** |
| **Child sex** | | | | |
| Male | 1.00 | - | 1.00 | - |
| Female | 0.95 | 0.86–1.05 | 0.93 | 0.84–1.03 |
| **Place of birth** | | | | |
| Urban | 1.00 | - | 1.00 | - |
| Rural | 1.68 | 1.49–1.90*** | 1.24 | 1.02–1.5* |
| **Maternal Age** | | | | |
| Adolescent (<20 years) | 4.36 | 3.7–5.14*** | 3.16 | 2.60–3.85 |
| 20–29 years | 1.00 | - | 1.00 | - |
| 30 years+ | 0.47 | 0.42–0.52*** | 0.49 | 0.43–0.55*** |
| **Education level** | | | | |
| None | 1.00 | - | 1.00 | - |
| Primary | 1.13 | 1.02–1.25** | 0.98 | 0.87–1.11 |
| Post primary | 0.70 | 0.62–0.80 | 0.81 | 0.69–0.96** |
| **Wealth index** | | | | |
| Index 1–2 | 1.00 | - | 1.00 | - |
| Index 3 | 1.04 | 0.92–1.18 | 0.89 | 0.77–1.02 |
| Index 4–5 | 0.58 | 0.51–0.66*** | 0.66 | 0.53–0.82*** |
| **Marital status** | | | | |
| Married | 1.00 | - | 1.00 | - |
| Unmarried | 1.33 | 1.14–1.55*** | 1.22 | 1.01–1.48** |

HR- Hazard Ratio

***P<0.001

**P<0.01.

system. The difference in the mortality estimates between HDSS and other surveys that are based on probability sampling such as DHS data are expected since the HDSS collected data on all registered household and thus, HDSS data may provide accurate and reliable estimates on life events in their regional localities. However, from the analysis of this data, we identified that there was a mismatch between birth registration and verbal autopsies data, where the two could not be linked as the birth registration unique identifier was missing in the verbal autopsy data. Yet matching the two datasets would provide the direct estimates of cause-specific mortality rates. Our analysis, therefore, contributes to the identification of data gaps for data quality and system strengthening strategies for existing and emerging HDSS.

The high proportion of antepartum morbidities that this study identified indicates that the absence of adequate prevention, identification, and treatment interventions predisposes women to the increased risks of adverse birth outcomes [39,40]. Additionally, health services' access failures or delays during labour contributes to the increased likelihood of new-born mortality or other injuries that may lead to long-term consequences. The substantial

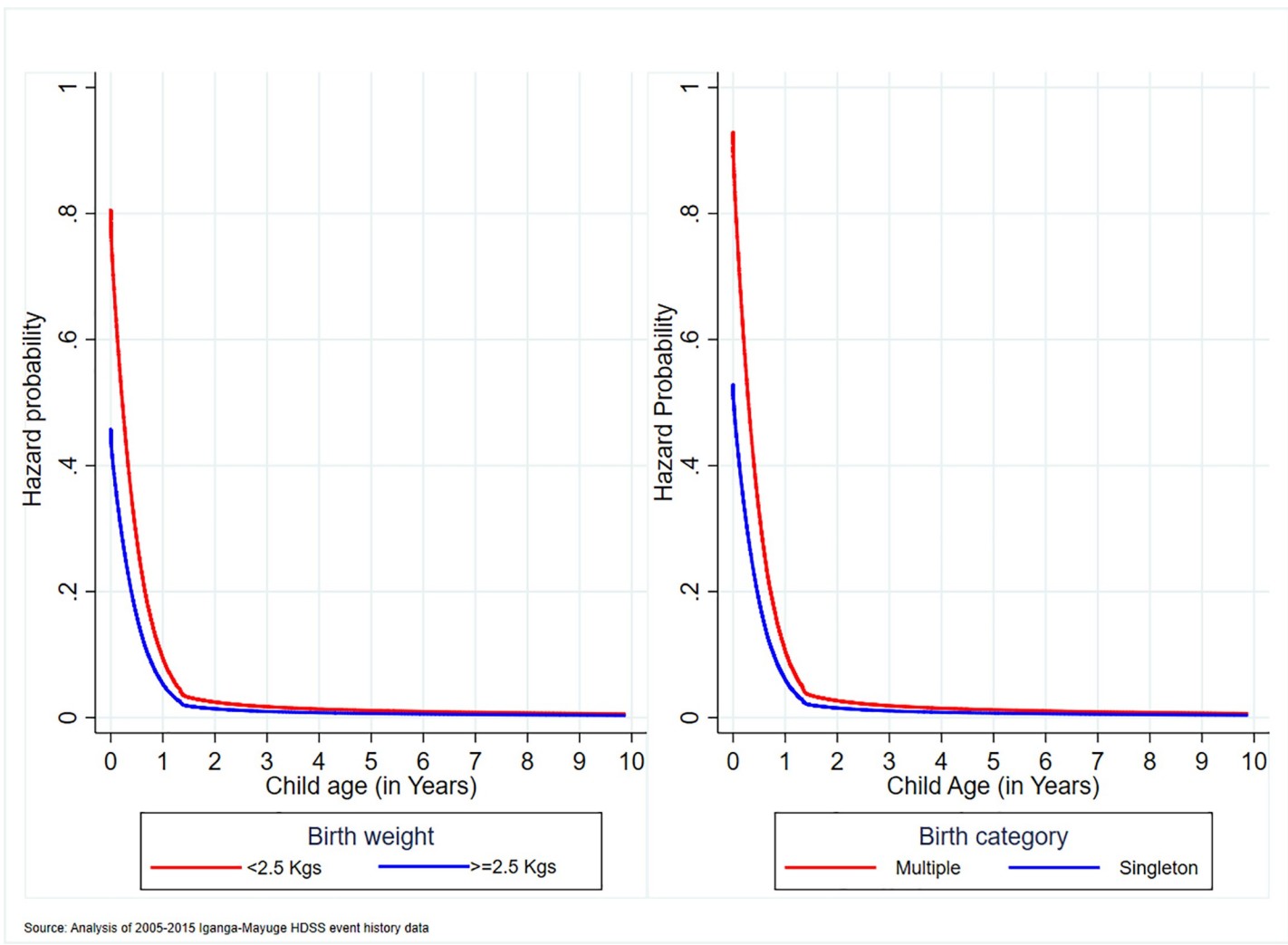

**Fig 3. Association of birth weight and birth category with under-10 mortality.**

proportion of deliveries conducted outside the health facility (30%) confirms the exiting challenges in accessing appropriate services since, in such low resources setting, the home deliveries are normally conducted under the assistance of unskilled health personnel. Indeed, the high proportion of stillbirth and birth asphyxia and injuries that this study has identified as major causes of mortality are related to the failures or delays in accessing appropriate healthcare services. In such context, the delays or failures in reaching the health facility are usually associated with transport challenges in accessing the health facility [41]. In fact, this study has indicated that 10% of the women who delivered in the community occurred while travelling to the health facility. Improved systems for prevention, identification and treatment of maternal morbidities at both community and health facility level in addition to improving access to required services is important in saving mothers with obstetric complication and new-born with danger signs such as birth asphyxia and LBW. Furthermore, improving referral transport systems and women birth preparedness practices could contribute to the improved timely access to the required maternal and new-born health services.

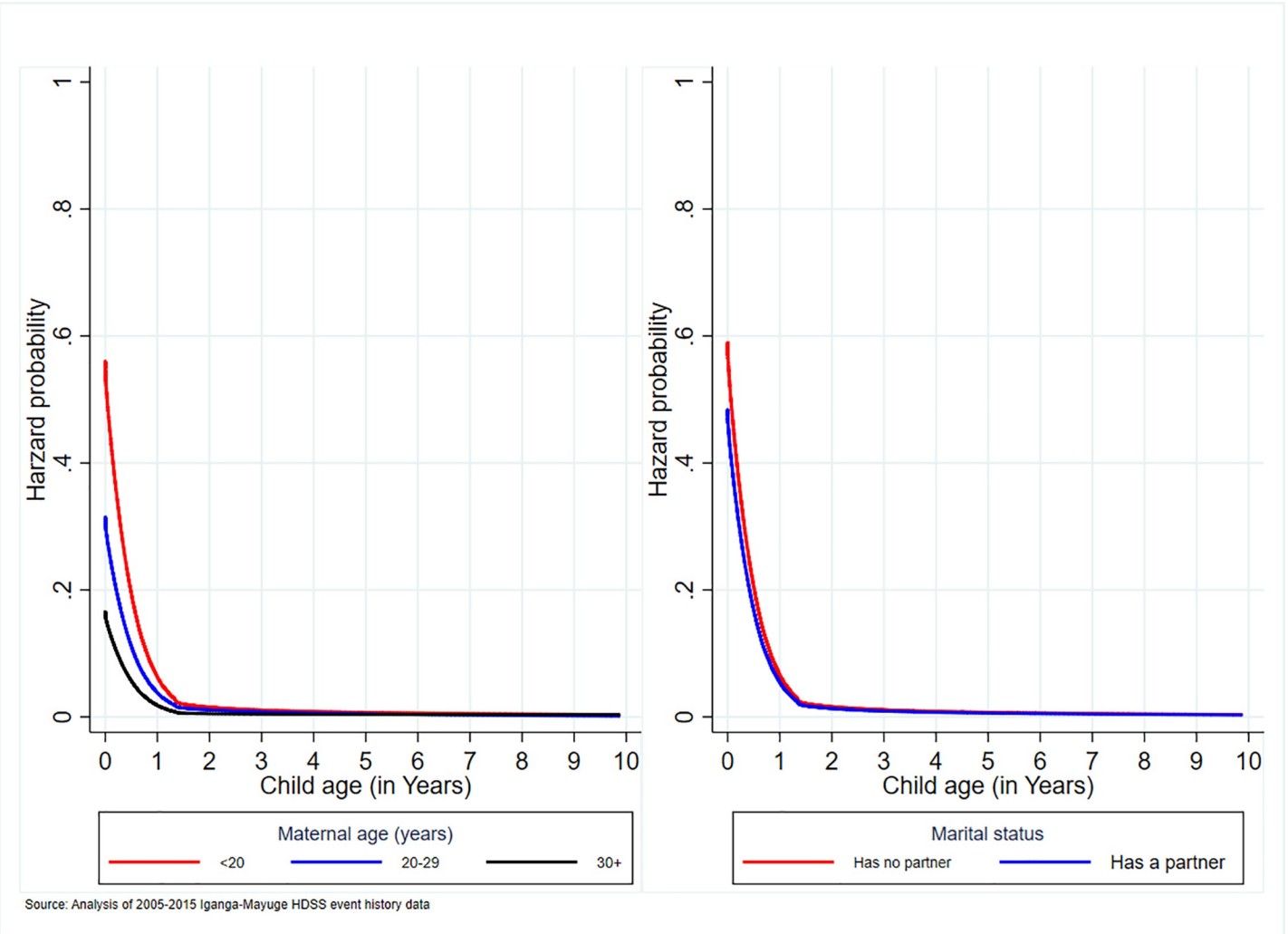

**Fig 4. Association of maternal age and marital status with under-10 mortality.**

The notable differences in causes of mortalities between children aged 0–5 and 5–9 years confirm an epidemiological shift of diseases between age bands. Malaria, protein deficiencies or malnutrition and acute respiratory infection (ARIs) were the major causes of mortality among children aged 0–5 years but these reduced by almost a half among those aged 5–9 years. Gastrointestinal disorder and injuries were substantial emerging morbidities after 4 years of age. These emerging morbidities could be resulting from the transition of new behaviours and curiosity among children aged 5–9 years and thus, their limited supervision may predispose them to unintended injuries such as poisoning, falls, drowning, burns and suffocation [42–44]. Similarly, children aged 5–9 years may be exposed to ingestion of unhealth items that may lead to gastrointestinal disorders such as worms and typhoid. The variation in some of the mortality causes across the age groups suggests age-specific interventions at different levels of growth. Morbidities like malaria, protein deficiency, diarrhoea and ARIs identified in this study as some of the major causes of mortality among children aged 5–9 years indicate the need for prevention and treatment interventions that target all age groups. However, access to the treatment and preventive interventions is still a challenge in SSA. For instance, in Uganda,

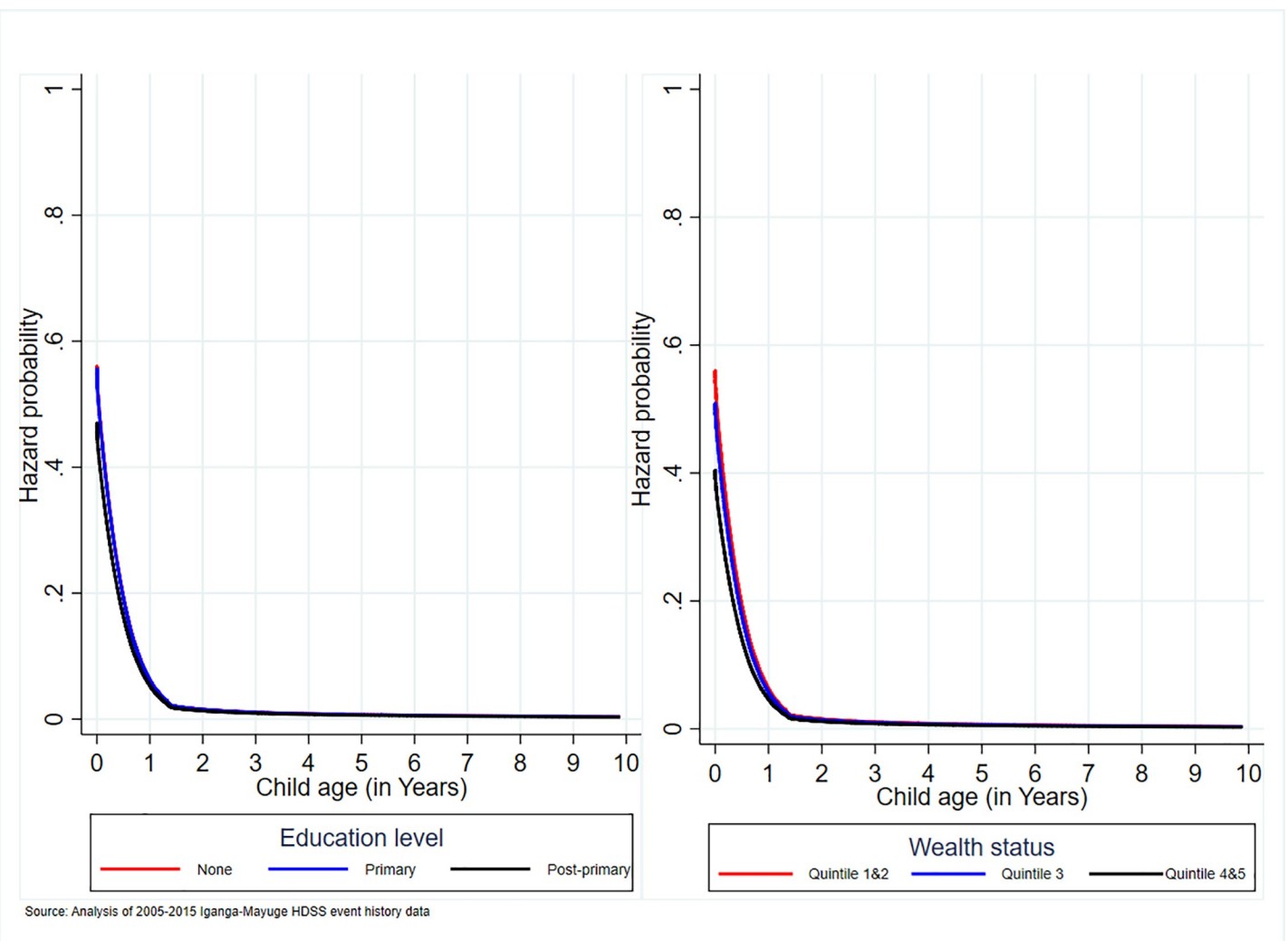

**Fig 5. Association of maternal education and wealth status with under-10 mortality.**

46% and 90% of children have no access to pneumococcal and rotavirus vaccines respectively [45] and such could be worse in rural communities. Further, despite the recommendation of accessing treatment within 12 hours for children with suspected symptoms of fever, malaria, cough, and diarrhoea, this study revealed that on average, most children with these diseases were taken to the health facility after 4 days of symptom recognition and 30% never reached the health facility at all. Childcare integrated community prevention package targeting parents and caretakers as well as interventions for early identification and treatment of morbidities are needed. Community interventions that have been effective in such low resource settings include integrated community case management [46,47] and use of community health work-ers for mobilisation, sensitisation and community-to-facility linkage[48,49].

Adolescence age and unmarried or single motherhood status were associated with increased risk of child mortality although the association was stronger among infants. Some of the worrying findings identified are that some girls become pregnant as early as 10 years and 26% of the adolescent mothers were aged 17 years and lower. Such indicates that a mother whose first birth was between 10 and 17 years would have a 7-year-old child and many more

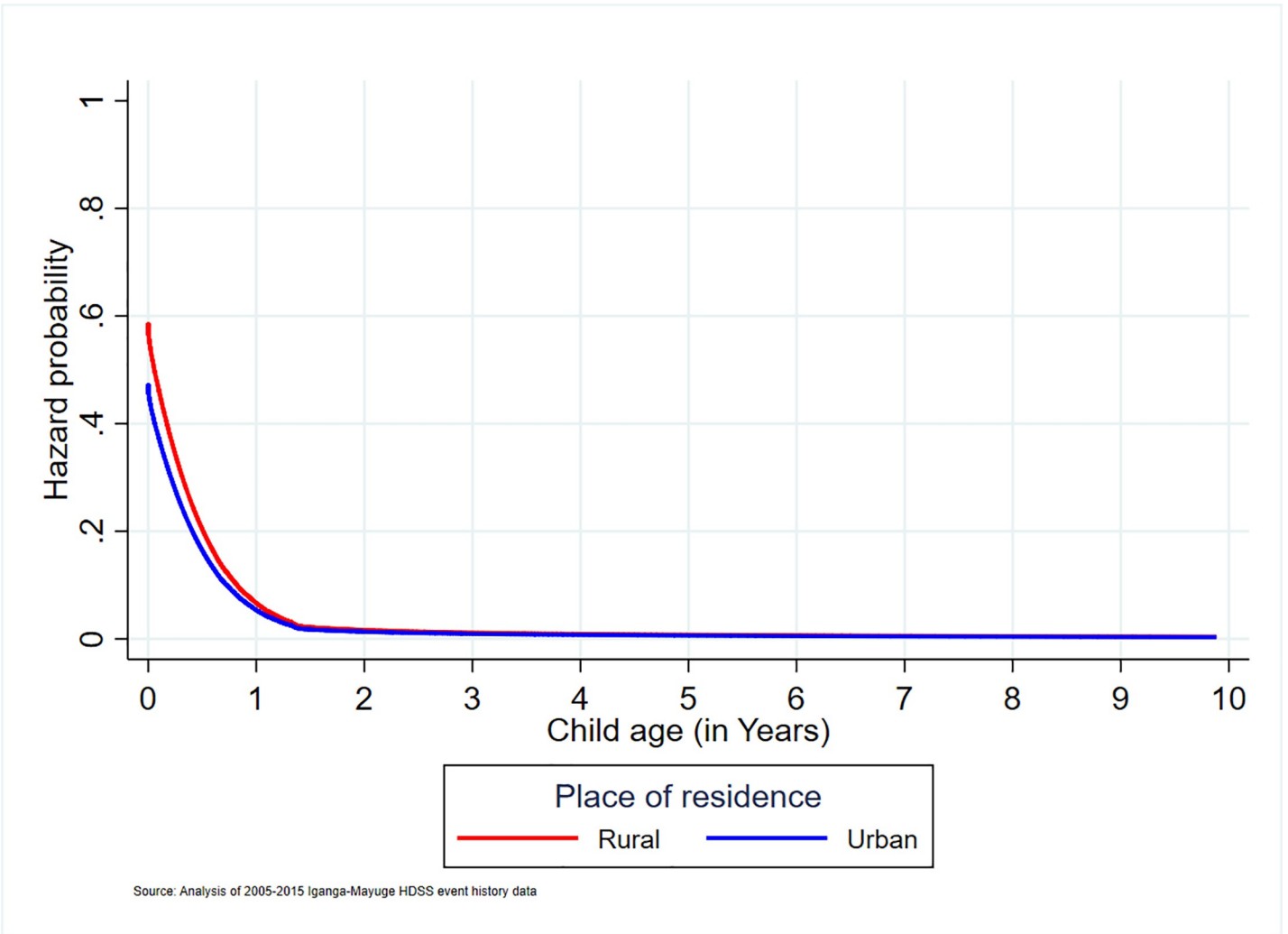

**Fig 6. Association of place of residence with under-10 mortality.**

children while still an adolescent or youth. Indeed, these mothers are characterised by other mortality risk factors such as LBW, low level of education, having no partner, lack of access to the resources. The adolescent women are normally isolated in the communities and not engaged in any income-generating activity and thus having inadequate support to cater for the health of their children. The effect of marital status could also be via access to income and decision-making power, which normally shapes the choice of healthcare services. Married women may have collective decision-making and support, which may contribute to child health and wellbeing. These results suggest interventions that address issues of teenage pregnancy in such settings. One of the suggested interventions though still controversial in Uganda's setting because of culture and religion, would be pushing for improved knowledge of and access to contraceptives methods, which could help reduce unwanted pregnancies. For teenagers that become pregnant or deliver, there is a need for interventions that help them produce and sustain healthy babies.

LBW and multiple births, which could be related, were important individual risk factors of child survival, particularly, the infants. The LBW babies are always susceptible to infections

and other risk factors such as proteins deficiency that may increase their risk of death [50,51]. Similarly, multiple births may be susceptible to malnutrition and ultimately infections, which would increase their risk of mortality. Worryingly, despite the importance of recording birth weight for identification of LBW babies, a large number of births had no birth weight taken and this has been indicated as a challenge in SSA countries[52,53]. Intervention targeting LBW risk factors such as maternal chronic illnesses, substance abuse, poor nutritional status, infections, maternal age that is under 20 and over 30, short birth intervals, and high parity that should be designed and implemented to reach women and girls. For instance, improved access to family planning has multiple effects on reducing LBW risk factors such as deliveries among high parity women, pregnancies following short birth intervals, and adolescent pregnancies. The other low cost and effective interventions that contribute to the identification and survival of LBW babies in resource-limited settings are foot length measurements [54–56] and Kangaroo Mother care [57–59] respectively. As alluded to earlier, it is also recommended that mothers attend ANC as required if they are to benefit from the interventions that prevent or control LBW effect and identify complications such as multiple births. Additionally, there is a need for guidelines on how to care for LBW and multiple births at the facility and community level [60]. Further, the effect of LBW and multiple births on the survival of children beyond the first month of life suggest for strategies that improve the survival of such groups beyond infancy.

Higher levels of education and belonging to the least poor household were inversely associated with the increased likelihood of child mortality but the association was higher among the infants. Nattey et al., 2013 indicate that child mortality is likely to be higher among mothers with no education level at all as compared to those with primary level and higher [61]. Girls from poorest families often drop out of school to complement on the household's income and ultimately get exposed to early sex as well as pregnancy [62,63] and such are common in LMICs [64]. Several studies have indicated the contribution of maternal education towards better health behaviours' practices such as nutrition, childhood immunization and better sanitation [65,66]. Regarding the wealth effect, poor groups within the communities are always at high risks of several morbidities and mortality [67] as such groups often lack access to quality of health services resulting from their limited decision-making power over the choice of healthcare.

The likelihood of mortality was also high among rural residents, but the association was higher among the infants. Noteworthy is that the socio-economic, maternal factors and health interventions that affect the child health may vary across and within regions, which perhaps could explain mortality variations [68,69]. The place of residence is characterised by community behaviours such as alcohol use, geographic characteristics such as calamities, facility proximity all which are well known to affect the health and survival of the child. In Uganda's rural context, the high child mortality rates in rural communities are not surprising as such communities are characterised by limited access to health facilities, poor transport systems, and poverty with each contributing to the delays or failure in accessing the required health services. As earlier mentioned, even the few health facilities in such communities frequently experience inadequate amenities and qualified health workers to provide the required services.

In conclusion, the mortality causes such as LBW, premature, birth asphyxia injuries, diarrhoea, malaria and pneumonia as well as some of the identified risk factors such as adolescent pregnancy and LBW in this study could be avoidable and amendable if girls/youths, pregnant women, women in labour and sick children received required prevention and treatment interventions. We also note that proteins deficiency or malnutrition is in most cases not recognised as a direct causes of child mortality and [23,70] and has not gained global and national attention, yet in such resource-limited context, the prevalence of malnutrition is high [70]. These results emphasize the need for a life-course [4,7,21,22,71] approach in the design and

implementation of child health interventions that include the pre-conception, pregnancy, birth or labour, and children of all age interventions. Some of the pre-conception and after delivery interventions include increased access to contraceptive use among adolescents and newly delivery women, which reduce adolescent pregnancy and short birth intervals. Pregnancy interventions may include strategies that mobilize women to attended antenatal care service including sensitization on care for pregnancies, which may contribute to the prevention of LBW and identification of pregnancy complications and morbidities at an early stage. For new-borns that are born with adverse outcomes such as LBW, preterm, and multiple births, they should be exposed to appropriate treatment and care interventions until 5 years of age. Such interventions should particularly target the uneducated, poor, and rural women dwellers. Other interventions on the prevention of accidents or injuries, hygiene-related infections and nutrition should target mothers, women, and other children caretakers. In addition to under-10 protein deficiency research studies, we also recommend for more evidence on how child morbidity and treatment access vary across different groups, which may guide the design of group-specific interventions.

## Strengths and limitations of the study

The strength of this study is that it used a combination of a decade of population longitudinal event history event histories and verbal autopsy data that have never been exploited. The HDSS data have been indicated to have the statistical power that is close to 100%, which gives them greater confidence in identifying significant differences [72]. This study has several limitations that should be acknowledged. First, the results could be generalised in the study area region, however, considering the fact that most of the communities not only in Uganda but also other Sub-Saharan Africa share the same community and health characteristics, such results could guide the design of health systems interventions that address both mortalities causes and risk factors. Second, the outmigration could have biased the mortality estimates; however, adjusting for censoring using life table analysis approach could have minimised the bias. Third, there was a considerate proportion of missingness for some of the variables, particularly birth weight that could lead to bias, but we minimised this through multiple imputations approach. Forth, the results on health care access at the time of delivery and for sick children are based on verbal autopsy data with no comparison, however, given the large record of verbal autopsy data, the analysis on when health care was accessed gives an insight into service's access behaviours and delays. Fifth, the causes of death are based on the child's mother narrative of the signs and symptoms recognised before death and the final codes assigned by the physicians, which may lead to the misclassification of the death causes; however, the two physicians that are employed by the HDSS review all the same verbal autopsy data independently to determine the causes and chain of event that lead to death, and in case of differences, the two meet to agree on the code. Lastly, this study was limited to the available data variables. For instance, information on pregnancy gestation period, which could have been used for LBW or premature and stillbirth classification was not available in the dataset this study used.

## Supporting information

**S1 File. The life-table analysis approach.**
(DOCX)

**S1 Table. Individual characteristics distribution for completed and imputed data using 2005–2015 birth registration dataset.**
(DOCX)

**S2 Table. Determinants of birth weight, wealth and marital status missingness in birth registration dataset.**
(DOCX)

**S3 Table. Maternal morbidities experienced in the last 3 months of pregnancy using 2005–2015 Iganga-Mayuge verbal autopsy data.**
(DOCX)

## Acknowledgments

We are grateful for the feedback provided by Bernardo Hernandez Prado and another anonymous reviewer who provided insightful suggestion during the peer review process. We also thank the Iganga-Mayuge HDSS data collection team and mothers within the HDSS catchment area.

## Author Contributions

**Conceptualization:** Rornald Muhumuza Kananura.

**Data curation:** Rornald Muhumuza Kananura.

**Formal analysis:** Rornald Muhumuza Kananura.

**Supervision:** Tiziana Leone, Arjan Gjonca.

**Validation:** Tiziana Leone, Dan Kajungu, Peter Waiswa.

**Writing – original draft:** Rornald Muhumuza Kananura.

**Writing – review & editing:** Rornald Muhumuza Kananura, Tiziana Leone, Tryphena Nareeba, Dan Kajungu, Peter Waiswa, Arjan Gjonca.

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
