## [Decision Letter · Decision Letter 0]

1 May 2020

PONE-D-20-06933

Under 10 mortality patterns, risk factors, and mechanisms in low resource settings: An analysis of event history demographic and verbal autopsy data

PLOS ONE

Dear Mr. Kananura,

Thank you for submitting your manuscript to PLOS ONE. After careful consideration, we feel that it has merit but does not fully meet PLOS ONE’s publication criteria as it currently stands. Therefore, we invite you to submit a revised version of the manuscript that addresses the points raised during the review process.

We would appreciate receiving your revised manuscript by Jun 15 2020 11:59PM. To enhance the reproducibility of your results, we recommend that if applicable you deposit your laboratory protocols in protocols.io, where a protocol can be assigned its own identifier (DOI) such that it can be cited independently in the future. For instructions see: http://journals.plos.org/plosone/s/submission-guidelines#loc-laboratory-protocols

We look forward to receiving your revised manuscript.

Kind regards,

Kannan Navaneetham, PhD

Academic Editor

PLOS ONE

Journal Requirements:

3. Please include a caption for figures 3-6

Reviewers' comments:

Reviewer's Responses to Questions

**Comments to the Author**

1. Is the manuscript technically sound, and do the data support the conclusions?

Reviewer #1: Yes

Reviewer #2: Yes

2. Has the statistical analysis been performed appropriately and rigorously? 

Reviewer #1: Yes

Reviewer #2: Yes

3. Have the authors made all data underlying the findings in their manuscript fully available?

Reviewer #1: Yes

Reviewer #2: Yes

4. Is the manuscript presented in an intelligible fashion and written in standard English?

Reviewer #1: Yes

Reviewer #2: Yes

5. Review Comments to the Author

Reviewer #1: This is an interesting paper exploring under 10 mortality patterns, risk factors and mechanisms in low resource settings in Uganda. The authors have done a careful analysis and methods and results are clearly presented. There is a good discussion of results and conclusions are supported by the results. However, there are some questions and suggestions I would have on my end:

1. The paper is part of a doctoral thesis work. It still includes some details in methods and results that are not usually reported in scientific paper, although they are crucial in a thesis. Compressing the paper would facility its dissemination and understanding. For example, the paper now has 7 tables and 6 figures, unusually long for a scientific paper. I would suggest to select only the key results for the body of the text and move some tables and figures to additional files. the methods section can be simplified as well.

2. The authors mentioned VAs were used in the study, although unfortunately it was not possible to link their results with survey data to explore risk factors by cause of death. Anyway, the authors mentioned the cause of death was defined from a physician review (2) of the VA. Didn´they consider the use of any of the automated methods available now (Insylico, SmartVA?) It would be good to have more details on the method of review of the VA in the methods sections. it is only briefly mentioned, and details are actually provided in the discussion.

3. The authors used multiplie imputations to deal with the high missingnes found in the database. In page 9, line 178, it says "The place of delivery... and maternal age were associated with the birth weight missigness- indicating a possibility of birth weight missingness at random". If missingness is associated with some variables, how come it can be at random? I don´t think so. In this case, did the imputation method used is suitable for non-random missingness?

4. As the authors mentioned, there are not that many studies of under 10 mortality. The authors make a good case showing differences in causes and levels of death in some ages within this group (e.g. under vs. over 5). However, they need to make a strong case of why it is important to analyze this group, and what is the contribution of this study. These points are mentioned in the discussion, but authors could elaborate more on this.

Reviewer #2: Please find attached the file, review comments to the Authors

Overall

The topic itself is very interesting and contains meaningful insight, which should to be shared with the global health community. The manuscript deserves to be published; however, it needs to be revised.

The authors seem to have included too much information, particularly in the results and discussion session. I suggest the authors selecting only one or two study aim(s) and align the results and discussion with the study aim(s).

In addition, I suggest them comparing existing studies relating to under-five child mortality. In particular, the majority of the previous studies have been focusing on under-five child deaths or child mortality. If the authors are selling the potential readers on the under-10 child mortality, they need to make the manuscript more attractive by comparing their findings with the existing studies on the under-five child deaths. Also, I suggest they add at least one paragraph explaining current global strategy to tackle child mortality by applying life-course approach since they are also emphasizing the importance of the approach. I introduced some references they might refer to as below.

6. PLOS authors have the option to publish the peer review history of their article (what does this mean?). If published, this will include your full peer review and any attached files.

Reviewer #1: Yes: BERNARDO HERNANDEZ PRADO

Reviewer #2: No

---

## [Author Response · Author response to Decision Letter 0]

14 May 2020

A detailed point-by-point response to the editor's and reviewers’ comments and the way they are addressed in the paper is indicated in the Cover letter and Responses to Reviewers attached Documents.

---

## [Decision Letter · Decision Letter 1]

29 May 2020

Under 10 mortality patterns, risk factors, and mechanisms in low resource settings of Eastern Uganda: an analysis of event history demographic and verbal social autopsy data

PONE-D-20-06933R1

Dear Dr. Kananura,

We are pleased to inform you that your manuscript has been judged scientifically suitable for publication and will be formally accepted for publication once it complies with all outstanding technical requirements.

With kind regards,

Kannan Navaneetham, PhD

Academic Editor

PLOS ONE

Additional Editor Comments (optional):

Reviewers' comments:

Reviewer's Responses to Questions

**Comments to the Author**

1. If the authors have adequately addressed your comments raised in a previous round of review and you feel that this manuscript is now acceptable for publication, you may indicate that here to bypass the “Comments to the Author” section, enter your conflict of interest statement in the “Confidential to Editor” section, and submit your "Accept" recommendation.

Reviewer #1: All comments have been addressed

Reviewer #2: All comments have been addressed

2. Is the manuscript technically sound, and do the data support the conclusions?

Reviewer #1: Yes

Reviewer #2: Yes

3. Has the statistical analysis been performed appropriately and rigorously? 

Reviewer #1: Yes

Reviewer #2: Yes

4. Have the authors made all data underlying the findings in their manuscript fully available?

Reviewer #1: Yes

Reviewer #2: Yes

5. Is the manuscript presented in an intelligible fashion and written in standard English?

Reviewer #1: Yes

Reviewer #2: Yes

6. Review Comments to the Author

Reviewer #1: The authors have satisfactorily addressed all the reviews. I have no further comments to this paper.

Reviewer #2: (No Response)

7. PLOS authors have the option to publish the peer review history of their article (what does this mean?). If published, this will include your full peer review and any attached files.

Reviewer #1: Yes: BERNARDO HERNANDEZ PRADO

Reviewer #2: No

---

## [Editor Report · Acceptance letter]

2 Jun 2020

PONE-D-20-06933R1 

Under 10 mortality patterns, risk factors, and mechanisms in low resource settings of Eastern Uganda: an analysis of event history demographic and verbal social autopsy data 

Dear Dr. Kananura:

I'm pleased to inform you that your manuscript has been deemed suitable for publication in PLOS ONE. Congratulations! Your manuscript is now with our production department. 

Kind regards, 

on behalf of

Professor Kannan Navaneetham 

Academic Editor

PLOS ONE